Functional flexibility in wild bonobo vocal behaviour

Clay Zanna 1 2 z.clay@bham.ac.uk
Archbold Jahmaira 2
Zuberbühler Klaus 2 3
1 School of Psychology, University of Birmingham , Birmingham , UK
2 Department of Comparative Cognition, Institute of Biology, University of Neuchatel , Neuchatel , Switzerland
3 School of Psychology & Neurosciences, University of St Andrews , St Andrews, Fife , UK
Vonk Jennifer
Electronic publication date: 2015 Aug 4
Publication date: 2015
Volume: 3
Electronic Location ID: e1124
Received 2015 May 9; Accepted 2015 Jul 3
Copyright: © 2015 Clay et al.
Copyright year: 2015
Copyright holder: Clay et al.
License: This is an open access article distributed under the terms of the Creative Commons Attribution License, which permits unrestricted use, distribution, reproduction and adaptation in any medium and for any purpose provided that it is properly attributed. For attribution, the original author(s), title, publication source (PeerJ) and either DOI or URL of the article must be cited.
License URL: https://creativecommons.org/licenses/by/4.0/

Keywords: Vocal development, Speech evolution, Great ape, Pre-linguistic infant, Emotion valence, Vocal flexibility, Primate, Language evolution, Protophone

Funding: L.S.B. Leakey Foundation National Geographic Society British Academy European Union Seventh Framework Programme 283871 This research was financially supported by the L.S.B. Leakey Foundation, the National Geographic Society: Committee for Research and Exploration Grant, the British Academy Small Research Grant, the European Union Seventh Framework Programme for research, technological development, and demonstration under grant agreement 283871 and private donors associated with the British Academy and the Leakey Foundation. The funders had no role in study design, data collection and analysis, decision to publish, or preparation of the manuscript.

==============================
A shared principle in the evolution of language and the development of speech is the emergence of functional flexibility, the capacity of vocal signals to express a range of emotional states independently of context and biological function. Functional flexibility has recently been demonstrated in the vocalisations of pre-linguistic human infants, which has been contrasted to the functionally fixed vocal behaviour of non-human primates. Here, we revisited the presumed chasm in functional flexibility between human and non-human primate vocal behaviour, with a study on our closest living primate relatives, the bonobo (Pan paniscus). We found that wild bonobos use a specific call type (the “peep”) across a range of contexts that cover the full valence range (positive-neutral-negative) in much of their daily activities, including feeding, travel, rest, aggression, alarm, nesting and grooming. Peeps were produced in functionally flexible ways in some contexts, but not others. Crucially, calls did not vary acoustically between neutral and positive contexts, suggesting that recipients take pragmatic information into account to make inferences about call meaning. In comparison, peeps during negative contexts were acoustically distinct. Our data suggest that the capacity for functional flexibility has evolutionary roots that predate the evolution of human speech. We interpret this evidence as an example of an evolutionary early transition away from fixed vocal signalling towards functional flexibility.

Introduction

A growing body of research suggests that human infant vocal development reveals something about the evolutionary history of language (Tomasello et al., 2005; Locke & Bogin, 2006; Tomasello, 2008). This is because a basic principle of evolution is that, similar to processes in development, natural selection acts by modifying or adding complexity to existing structures and mechanisms rather than by generating entirely new ones, a logic that has also been applied to the evolution of language (Fitch, 2010). A research goal therefore is to describe the basic design principles of early stages of vocal behaviour, which may have served as building blocks on which subsequent stages of linguistic development have emerged (Oller, 2000; Tomasello et al., 2005; Oller et al., 2013). One of these building blocks is ‘functional flexibility’, an individual’s capacity to produce signals that are detached from a predetermined function to express different psychological and affective states in a range of situations (Griebel & Oller, 2008; Oller et al., 2013). This is thought to contrast with animal signals and some human vocalisations (e.g., crying, laughter), which are tightly linked to specific psychological and motivational states (Oller et al., 2013).

Part of this reasoning stems from a paucity of evidence of functionally flexible signalling in non-human primates (Oller et al., 2013; Ackermann, Hage & Ziegler, 2014 but see Lameira et al., 2013). In contrast, recent analyses of prelinguistic human infant vocal interactions showed that three types of ‘protophone’ vocalisations were used flexibly to express a full range of emotional content (positive, neutral and negative valence) across a range of different situations (Oller et al., 2013). Moreover, some vocal types (squeals, vowel-like sounds and growls) were associated with neutral facial expressions, further suggesting that, in human infants, vocal production can be detached from a specific biological function. This is in contrast to other infant vocalisations, namely crying and laughter, which appear to be fixed to specific affective states. It has therefore been suggested that these types of affect-bound, or functionally fixed, vocalisations resemble non-human primate calls, both in terms of their form and function, as well as the brain mechanisms underlying them (Newman, 1985; Owren, Amoss & Rendall, 2011; Bryant & Aktipis, 2014; Owren & Amoss, 2014).

In animals, functionally fixed signals are often considered equivalent to ‘functionally referential’ signals, broadly defined as acoustically distinct signals produced in response to a narrow range of stimuli to the extent that a receiver, upon hearing the signal in the absence of the stimuli, responds as if experiencing the eliciting stimuli itself (Macedonia & Evans, 1993). The alarm calls of numerous primates, for example, have been discussed as being functionally fixed in that they are produced reliably in response to certain classes of predators (such as aerial alarm calls in response to aerial predators) and their underlying affective states (Seyfarth, Cheney & Marler, 1980a; Seyfarth, Cheney & Marler, 1980b; Zuberbühler, 2003; Zuberbühler, 2006). While some calls may fit the notion of functional fixedness, the theoretical assumption that vocalisations must be produced to a narrow range of stimuli in order to functionally refer to something in the world has, in fact, been recently been challenged on the grounds that calls produced to a number of different stimuli may still be functionally referential, given the availability of other contextual cues (Scarantino & Clay, 2014).

More recent research on animal vocal signalling, particularly by non-human primates, is increasingly suggesting scope for greater flexibility; the spontaneous production of unvoiced, atypical calls by captive great apes being one such example. In many captive facilities, there have been reports of individuals producing atypical, voiceless calls (e.g., “raspberries”) in order to gain the attention of human caregiver, and that some of these are acquired through social learning (e.g., Gorilla gorilla gorilla (Perlman, Patterson & Cohn, 2012); Pongo pygmaeus (Wich et al., 2009; Hardus et al., 2009; Lameira et al., 2015; Pan troglodytes (Hopkins, Taglialatela & Leavens, 2007; Taglialatela et al., 2012); Pan paniscus (Taglialatela & Savage-Rumbaugh, 2003)). Moreover, conditioning experiments with rhesus macaques and cotton-top tamarins have highlighted considerable vocal control, with individuals able to initiate vocalisations and modify various features of their vocal output in response to different external stimuli (Hage, Gavrilov & Nieder, 2013; Hotchkin, Parks & Weiss, 2013). Studies of natural communication among conspecifics have also highlighted considerable vocal flexibility. For instance, studies in the contexts of travel recruitment (Gruber & Zuberbühler, 2013), feeding advertisement (Slocombe et al., 2010; Schel et al., 2013a; Schel et al., 2013b), sexual interactions (Townsend, Deschner & Zuberbühler, 2008), social greetings (Laporte & Zuberbühler, 2010) and predator alarms (Crockford et al., 2012; Schel et al., 2013a; Schel et al., 2013b) have revealed that wild chimpanzees have a notable degree of control over call production and flexibly modify it in response to different audience compositions and, in some cases, in intentional ways (Crockford et al., 2012; Schel et al., 2013a; Schel et al., 2013b). Group-specific ‘dialects’ have also been suggested for four chimpanzee communities living in the same forest in the Ivory Coast (Crockford et al., 2004). In orang-utans, a recent study of the production of variants of an alarm call (the “kiss-squeak”) in two different wild populations revealed population-specific usage, which indicated a certain degree of arbitrariness (Lameira et al., 2013).

Research on the communicative behaviour of a human-enculturated and language-trained bonobo (‘Kanzi’) has already highlighted bonobos as an interesting candidate species to examine flexible primate vocal production (Savage-Rumbaugh & Lewin, 1994). It has been suggested, for instance, that Kanzi used ‘peeps,’ one of the most common vocalisations in the bonobo repertoire (de Waal, 1988), to communicate with caregivers in ways that conform to conversational rules used in human speech dialogue (Greenfield & Savage-Rumbaugh, 1990; Greenfield & Savage-Rumbaugh, 1993; Savage-Rumbaugh, 1990; Savage-Rumbaugh, 1988; Segerdahl, Fields & Savage-Rumbaugh, 2005). Another study also suggested that Kanzi’s vocal repertoire had become augmented to include novel acoustic variants of the ‘peep’ vocalisation produced in response to different food types and that, astonishingly, the acoustic structure of these vocal variants resembled the corresponding spoken English words used by his caregivers (Hopkins & Savage-Rumbaugh, 1991; Taglialatela & Savage-Rumbaugh, 2003).

While Kanzi’s communicative behaviour is suggestive of considerable communicative flexibility, research into flexibility in the natural vocal communication system of bonobos remains scant (Liebal et al., 2013). The bonobo vocal repertoire has been described as highly graded, containing up to fifteen principal call types (de Waal, 1988; Bermejo & Omedes, 1999). Consistent with the more traditional view of primate vocal signalling, most bonobo call types appear to be tied to particular emotional states or valence classes, such as pant-laughing during socio-positive interactions, screaming and pout-moaning in response to agonism, threat barks during aggressive attacks, pant-grunts during submissive greetings and alarm barks in response to predators (de Waal, 1988; Z Clay, L Ravaux, FBM de Waal, K Zuberbühler, 2015, unpublished data). However, as indicated in the Kanzi studies, their most frequent vocal type, the ‘peep,’ is produced across an array of behavioural contexts (Fig. 1, de Waal, 1988). Analyses of the bonobo vocal repertoire in the wild (Bermejo & Omedes, 1999) and in captivity (de Waal, 1988) both stressed the importance of the peep in bonobo communication and reported its highly flexible and varied use across different behavioural contexts.

Figure 1 Time–frequency spectrograms illustrating peeps produced by four wild bonobos (BE, EM, ZD, male; NI, female) during different behavioural contexts.

The emotional valence of the context is indicated in parantheses.

Although peeps are produced across an array of contexts, they are especially common during feeding events and, consequently, have so far only been systematically studied as a food-associated signal (Van Krunkelsven et al., 1996; Clay & Zuberbühler, 2009; Clay & Zuberbühler, 2011). For instance, peeps produced in response to food discovery have been shown to be frequently combined with other call types into longer sequences, whereby the probabilistic organisation of the sequence conveys information to receivers about perceived food quality (Clay & Zuberbühler, 2009; Clay & Zuberbühler, 2011).

The apparently varied usage of the peep vocalisation in both natural and artificial settings suggests it to be an interesting candidate to study evidence for vocal flexibility. Moreover, promising findings from studies of Kanzi indicated that peeps may even have the potential to be used in language-like ways, such as conforming to conversational rules (Greenfield & Savage-Rumbaugh, 1990; Greenfield & Savage-Rumbaugh, 1993; Savage-Rumbaugh, 1990; Savage-Rumbaugh, 1988; Segerdahl, Fields & Savage-Rumbaugh, 2005) and being modified into context-specific acoustic variants that correspond to spoken human words (Taglialatela & Savage-Rumbaugh, 2003). Despite converging reports indicating flexibility in this vocalisation, to our knowledge, no systematic investigation of their general use has been conducted. For example, it is currently unknown whether peeps share the same acoustic structure across contexts or whether they are context-specific. Before investigating the degree to which this candidate call type may be produced flexibly, it is important to first establish whether their acoustic structure varies across contexts.

The aim of the current study is therefore to systematically analyse the acoustic structure of peeps to assess whether they are tied to specific behavioural contexts or whether, like human infants (Oller et al., 2013), bonobos are capable of producing the same vocalisation across a range of affective states. From an evolutionary perspective, looking for non-human primate vocalisations that are not so tightly tied to biological function but are used in more functionally flexible ways provides potentially relevant insights for the evolution of human speech. If humans are the only species capable of functional flexibility, that is, to produce the same vocalisation across different valence states, peeps should be expected to vary according to the valence (i.e., from positive to neutral to negative) of the contexts. To this end, we analysed various acoustic parameters of bonobo peeps produced in different behavioural contexts relating to the three principal valence dimensions (positive-neutral-negative) to explore whether these structural parameters provided acoustic cues relating to the inferred affective valence. This approach differs from the ‘discrete emotion’ approach, which presupposes discrete emotional states in animals and humans that arise in response to anticipation of rewarding or punishing events, such as fear or pleasure (Russell, 1980; Mendl, Burman & Paul, 2010). By taking a more graded approach, the current study is better suited for making direct comparisons with previous research on prelinguistic human infant vocalisations (Oller et al., 2013).

Materials & Methods

Behavioural observations were conducted on individuals from the Bompusa community of wild bonobos by ZC from October 2013 to March 2014 at Lui Kotale (managed by the Max Planck Institute for Evolutionary Anthropology, Leipzig) located near the Salonga National Park, in DR Congo. At this time, the fully habituated and fully identified community consisted of twelve adult females, two subadult females, five adult males two subadult males and eighteen immatures (juveniles and infants). Animal focal animal sampling (15 min) was conducted on all males and all adult females throughout the study period, amounting to an average of 11 focal hours per individual.

We recorded vocalisations produced by focal individuals across a variety of behavioural contexts, as they occurred. We recorded vocalizations at distances of 7–20 m using a Sennheiser MKH816T directional microphone and Marantz PMD661 solid-state recorder (Microphone frequency response: 50–20,000 Hz, ±3.5 dB; sampling rate of 44.1 kHz, 16 bits accuracy).

Our acoustic analyses focussed on the bonobo peep, a high-frequency, closed mouth vocalisation (approx. 2,200 Hz, (de Waal, 1988), see Fig. 1), short in duration (approx. 0.1 sec (de Waal, 1988; Clay & Zuberbühler, 2009)) and characterised by a simple, flat acoustic form composed of several harmonics that are generally un-modulated. In order to analyse the peep structure across different contexts, we first identified the most vocally active individuals from focal recordings, identifying those that produced vocalisations in at least two feeding contexts and two non-feeding contexts, which resulted in a sample of eight individuals (four adult males, one subadult male and three adult females). Behavioural contexts were mutually exclusive, i.e., peeps produced holding or consuming food while travelling or resting were excluded. In order to compare the acoustic structure of peeps in different contexts we compared the acoustic structure of peeps during the contexts that generated the most peep vocalisations per individual. We collected peeps produced at the onset of each behavioural context (i.e., at the onset of food discovery or travel). Because the number of peeps produced at the vocal sequence onset varied across different calling events, we analysed up the first three consecutive peeps produced at the beginning of a vocal event by the same individual as this was the typically number of peeps produced in a consecutive sequence. We calculated mean scores per parameter across the three peeps to standardise across the same calling event.

In contrast to an analysis focussing on discrete emotional states, we were interested in first establishing whether bonobo vocalisations may be used flexibly different valence contexts (positive-neutral-negative), as has been demonstrated in prelinguistic infants (Oller et al., 2013). Therefore, for each individual, we randomly selected a balanced sample of eight peep recordings produced during feeding contexts (feeding on shoots/seeds on the ground and fruits in trees), which we inferred to be as approximately positive in overall valence (Briefer, Tettamanti & McElligott, 2015) and eight peep events produced during non-feeding contexts (resting and travel), which we inferred to be, in comparison to feeding, relatively neutral in order to valence. This amounted to a total of 128 peep events. In order to capture the spectrum of emotional valence in our acoustic analyses (i.e., positive-negative-neutral), we also analysed a sample of peeps associated with predator alarm responses and in response to agonistic interactions as the victim, which were both taken to represent negative valence. As peeps in response to agonistic and alarm contexts were rare, we analysed a balanced and randomized sample of 4 peep samples per individual, taken from independent behavioural events produced by 7 of the original 8 individuals (N = 28 in total). The eighth individual was excluded in this sample due to inadequate sample size. We selected two contexts per valence class (positive-negative-neutral) in order to maximise sample size as well as to adequately capture the potential acoustic variation in different contexts. In order to capture variation in the feeding experience overall, recordings from feeding contexts included a randomized and balanced selection of vocal events in response to feeding on fruits in trees as well as to herbaceous shoots on the ground. For non-feeding contexts, we analysed a randomized sample of recordings for each individual produced during rest and travel on the ground. For negative valence contexts, we analysed a randomised balanced sample of peeps produced during agonistic conflicts and predator alarm contexts.

We carried out all quantitative acoustic analyses with Praat 5.4.01 using the following settings: analysis window length 0.05 s, dynamic range 70 dB; pitch range 500–3,000 Hz, optimized for voice analysis, spectrogram view range 0–10 kHz. We performed pitch analysis using a script (“Analyse Source Editor”) written by M Owren (pers. comm., 2007). We then took the following spectral measurements from the fundamental frequency (F0): (1) mean fundamental frequency (Hz): average F0 across the entire call; (2) frequency at call onset, (3) frequency at call middle; (4) frequency at call offset; (5) transition onset (Hz): frequency of maximum energy at call onset minus frequency of maximum energy at call middle; (6) transition offset (Hz): frequency of maximum energy at call middle minus frequency of maximum energy at call offset; (7) maximum fundamental frequency (Hz): maximum frequency of F0; (8) minimum fundamental frequency (Hz): minimum frequency of F0; (9) number of harmonics: number of harmonic bands visible. In the temporal domain, we measured the call duration (10).

Next, we screened the data for outliers by producing standardized Z scores, rejecting any calls with a Z score greater than 3.29 in one or more parameters (Tabachnick & Fidell, 2001). We regressed all parameters to check for multi-colinearity and singularity, removing parameters with a variance inflation factor greater than 10. We then conducted a Discriminant Function Analysis (DFA) to assess whether the uncorrelated acoustic variables could discriminate between different behavioural contexts. Each of the eight individuals equally contributed eight randomly selected calls for both food (henceforth ‘positive valence’) and non-food (henceforth ‘neutral valence’) contexts and four calls per individual were entered for the negative valence (N = 156 peep samples in total). To cross-validate the discriminant functions produced in the analysis, we used the leave-one-out classification procedure, which classifies each calls by the functions derived from all calls other than that one. We used Binomial tests to analyse whether the proportion of correct discrimination differed significantly from chance.

In order to examine whether peeps conveyed information about caller identity, we conducted a DFA using the same data used for the above analysis but taking individual identity as the discriminating factor. We additionally conducted separate DFAs for the positive and neutral valence contexts in order to control for behavioural context. We were unable to include separate DFAs for the negative valence context due to small sample size (N = 4 calls per individual) compared to the number of acoustic parameters under scrutiny, which led to inadequate statistical power.

Since the acoustic data were two-factorial (caller ID; context), it has been argued that conventional DFA does not allow for a valid estimation of the overall significance of discriminability (Mundry & Sommer, 2007). Therefore, for any significant DFA discrimination, we conducted a permuted Discriminant Function Analysis (pDFA), using a macro written by (Mundry & Sommer, 2007; R Mundry, pers. comm., 2007). The pDFA estimates the significance of the number of correctly classified calls (cross-validated), taking into account repeated contributions per individual caller.

Following significant discrimination in the pDFA and diagnostic tests, we used Univariate Analysis of Variance tests to explore whether each of the acoustic parameters varied statistically with context, entering Caller Identity as a Random Factor and Context as the Fixed Factor.

All statistical tests were carried out using SPSS version 21.0 (SPSS Inc., Cary, North Carolina, U.S.A.) and R Studio version 3.1.1 (The R Foundation for Statistical Computing, Vienna, Austria). All tests were two tailed and alpha levels were set at 0.05, unless stated as being corrected. We applied standardised Bonferroni corrections for multiple comparisons.

Results

During focal animal sampling, we recorded peeps in response to over a dozen different behavioural contexts, which, across all focal individuals, included feeding on fruits, leaves, seeds, flowers in trees and on shoots, seeds, leaves and fruits on the ground. It also included travelling, resting, grooming, preparing a nest, interacting sexually, responding to vocalisations from other parties, descending from trees after feeding, alarm responses to predators or unexpected events, weather changes, agonistic interactions, submissive or appeasement responses towards more dominant individuals, and vocal greetings to the arrival of another individual joining the party.

Acoustic structure of peeps

We compared the acoustic structure of peeps produced in different contexts (feeding; travel/rest; agonism/alarm) that were associated with different emotion valences (positive; neutral; negative valence, respectively), Fig. 1 and that generated the most peep vocalisations across individuals. Following a multi-colinearity screening, we entered six of the nine original acoustic parameters into our acoustic analyses for eight individuals (total N call events = 156: call duration, mean F0, F0 at call onset, number of harmonics, transition onset and transition offset) and applied logarithmic transformations on three of the acoustic parameters to improve their homogeneity of variance. Results from a cross-validated discriminant function analysis revealed that while the DFA model generated two significant discriminant functions (Wilks Lambda: 0.550, χ2 (df = 14) =80.007, P < .001), peeps produced in association with positive valence contexts could not be reliably discriminated from those produced in all other contexts: the functions only classified 49.3% of the calls correctly, which was below chance level (Binomial test (0.14) P > 0.05).

On a pairwise basis, DFA analyses further revealed that peeps produced in association with positive valence contexts could not be reliably discriminated from those produced during neutral valence contexts (Wilks Lambda: 0.947, χ2 (df = 6) = 6.638, P = 0.356). In a cross-validated analysis, the functions only classified 52.3% of the calls correctly, which was below chance level (Binomial test (0.5) P > 0.05). However, peeps associated with negative valence (i.e., alarm and agonism) could be significantly discriminated from those produced in association with positive valence (feeding) (82.1% of calls correctly classified; Wilks lambda = 0.468, χ2 (df = 7) = 59.602, P < .001; Binomial test (0.5) P < .001, Bonferroni corrections), which was validated in a subsequent pDFA controlling for repeated contributions (P = 0.009). Similarly, there was significant discrimination of peeps produced in response to negative valence contexts to those during neutral valence contexts, with 77.4% of calls (cross-validated) correctly classified (Wilks Lambda = 0.551, χ2 (df = 6) = 47.107, P < .001; Binomial test (0.5) P < .001, Bonferroni corrections).

Caller identity

We used the same cross-validated DFA procedure to test whether peeps could be acoustically discriminated on the basis of caller identity (N = 8 individuals). The model generated six significant discriminant functions (Wilks Lambda: 0.371, χ2 (df = 42) = 119.043, P < .001), which discriminated caller identity at a significantly higher rate than chance (cross-validated correct classification: 31.3%, Binomial test (0.125) P < 0.001).

We then conducted two separate DFAs to examine individual identity discrimination for peeps in positive and neutral contexts. Results from the analyses were equivalent, with identity significantly discriminated in both contexts (Individual identity in Feeding contexts 31.3% (20/64) calls correctly classified: Wilks Lambda = 0.234, χ2 (df = 42) = 81.285, P < 0.001; Binomial test (0.125) p < 0.001; in non-feeding contexts: 32.8% (21/64) calls correctly classified Wilks lambda = 0.210, χ2 (df = 42) = 87.313, P < 0.001; Binomial test (0.125) P < .001).

Comparing acoustic parameters

At the level of acoustic parameters, Univariate ANOVAs (Caller Identity as a random factor) revealed that the mean call duration, the mean fundamental frequency and the mean frequency at call onset varied significantly as a function of behavioural context (Mean call duration F2,12 = 5.625, P = 0.019; Mean F0: F2,12 = 19.054, P < .001; F0 call onset: F2,12 = 40.259, P < 0.001). Pair-wise comparisons (standard Bonferroni corrections), as shown in Fig. 2, of fundamental frequency (F0) parameters showed that peeps produced in association with negative valence had a significantly higher mean F0 and a higher onset F0 compared to peeps associated with positive valence (Mean F0negative = 2,131 ± 267 Hz, Mean F0positive = 1,660 ± 133; F1,6 = 16.862, P = 0.006; F0 at call onsetnegative = 2,027 ± 194 Hz, F0 at call onsetpositive = 1,612 ± 125 Hz; F1,6, = 35.990, P = 0.001) and neutral valence (Mean F0neutral = 1,584 ± 210 Hz: F1,6 = 27.160, p = 0.002; F0 at call onsetneutral 1,508 ± 186 Hz, F1,6 = 69.887, P < .001). Although peeps associated with negative valence were shorter in duration compared to those associated with positive valence (Mean call durationnegative = 0.12 ± 0.14, mean call durationpositive = 0.15 ± 0.03: F1,6 = 8.316, P = 0.028), the result was not significant under the Bonferroni correction. There were no other significant acoustic differences.

Figure 2 Boxplots indicating six acoustic parameters of peep vocalisations that varied as a function of behavioural context.

The emotional valence associated with the context is indicated in parentheses. Thick black lines represent medians; open circles and small asterisks represent outliers, box edges represent the upper and lower hinges of the H-spread, which generally matches the upper and lower quartiles; whiskers represent the adjacent values, which are the most extreme values still lying within hinges and the normal distribution of the sample. For significant differences, lines with ∗∗ represents P < .05, ∗∗∗ represents P < .001.

Discussion

Results suggest that, contrary to current models, humans may not be unique among primates in their ability to produce functionally flexible vocalisations. Our acoustic analyses suggest that the ‘peep’ calls of wild bonobos are produced in flexible ways in response to a range of different behavioural contexts of varying affective valence. Following similar evidence in pre-linguistic human infants (Oller et al., 2013), bonobos produced peep vocalisations across a range of neutral, negative and positive contexts in different behavioural situations. Although peeps produced in negative contexts differed acoustically, the acoustic structure of peeps produced in association with positive valence (feeding), could not be discriminated from neutral valence (travel, rest), despite the fact that behavioural contexts were mutually exclusive. Specifically, peeps produced in association with negative valence (alarm and agonism) possessed significantly higher mean fundamental frequencies, higher frequencies at call onset and shorter durations. The finding of acoustic variants within the same call type across different contexts is a relatively common finding in studies of primate vocal behaviour (Owren, Seyfarth & Cheney, 1997; Rendall et al., 1999; Rendall, 2003; Slocombe & Zuberbühler, 2007). Moreover, the fact that context-specific acoustic cues were found in negative but not other contexts has interesting implications both at the level of call production and evolutionary function. In terms of call production, the acoustic differences (higher frequencies and shorter call duration) found for negative contexts are most likely a direct result of higher subglottal air pressure during call production (Fitch & Hauser, 1995; Fitch, 2006), probably because individuals perceive situations as charged, tense and more urgent. This is likely to have direct physiological consequences such that the same volume of air is passing through the vocal tract but at a higher speed so that the vocal folds oscillate at a higher frequency for a shorter period of time.

The fact that peeps produced during negative contexts are more constrained by the physical mechanics of vocal production raises the possibility that, in the course of language evolution, functional flexibility may first have occurred in positive and neutral contexts. Generally, cues to emotional valence may be harder to conceal during negative contexts, which impedes the vocal control that is required for functional flexibility. Currently, it is not known whether the functionally flexible calls of human infants also vary acoustically as a function of affect valence, as no acoustic analyses have been conducted (Oller et al., 2013). Our prediction is that, even in humans, negative emotional valence is equally conveyed in the acoustic structure of vocalisations, including in human infant vocalisations.

Overall, our results suggest that bonobo peep production may represent a somewhat intermediate stage between functionally fixed (as seen for most primate vocalisations) and functionally flexible signals, as seen in most human vocalisations (Oller et al., 2013). Nevertheless, peeps could be reliably discriminated on the basis of caller identity across contexts, suggesting that the calls reliably convey other relevant information to receivers.

In order to make comparisons with evidence from pre-linguistic human infants (Oller et al., 2013), our study took a ‘dimensional approach’ to emotions (Russell, 1980; Watson et al., 1999; Mendl, Burman & Paul, 2010) by focussing on the valence associated with the eliciting behavioural contexts. According to the ‘dimensional approach,’ emotions are characterized along two core dimensions: valence (negative or positive; e.g., sad versus happy; Russell, 1980) and arousal (e.g., calm versus excited (Mendl, Burman & Paul, 2010). This contrasts with the ‘discrete emotions’ approach (Ekman, 1992; Panksepp, 1998), which although valuable for examining specific emotions, has been suggested to over-focus on certain emotions and not others (e.g., positive ones (Boissy et al., 2007), as well as lacking an overall framework that integrates a wider range of emotional states (Mendl, Burman & Paul, 2010). The ‘dimensional approach’ has been shown to be more powerful for studying animal emotions (Mendl et al., 2009; Mendl, Burman & Paul, 2010) across behavioural (Briefer, Tettamanti & McElligott, 2015; Reefmann, Wechsler & Gygax, 2009; Imfeld-Mueller et al., 2011), physiological (Da Costa et al., 2004; Davies, Radford & Nicol, 2014), and cognitive domains (Nettle & Bateson, 2012; Briefer & McElligott, 2013). Consequently, the current study is therefore unable to provide more detailed insights into the relationship between vocal production and discrete emotional states, such as fear or pleasure (Ekman, 1992; Panksepp, 1998). It can be assumed that considerable variation in discrete emotional states most likely exists within the contexts recorded here, for instance, feeding may elicit different discrete emotional states according to other related factors (e.g., social rank, age, group size). A substantial follow-up study, involving a greater sample size of both calls and contexts, that also controls for potentially confounding factors, would be needed to address the vocal indicators of discrete emotional states.

It has been suggested that the presence of functionally flexible vocalisations in pre-linguistic human infants is evidence for an evolutionary divergence towards speech production and variation in the expression of emotion across different utterances that sets humans apart from the rest of the primate lineage (Oller et al., 2013). For instance, these vocalisations, known as ‘protophones’ are produced across a full range of affect states and were the most commonly occurring vocal type, suggesting that even pre-linguistic infants possess considerable ‘exploratory vocal freedom’ and utilise a communication system that is predominantly detached from function. Protophones were most commonly associated with neutral facial expressions compared to cry and laughter, which were tightly linked to negative and positive affect, respectively, further highlighting their greater emotional detachment. Interestingly, bonobo peeps are also typically produced with a neutral facial expression, produced while the mouth remains closed without any particular facial expression (de Waal, 1988).

As always, it is somewhat problematic to draw firm conclusions from negative evidence, that is, the fact that we found no acoustic differences across non-negative valence contexts. For example, it is possible that there was subtle variation in other acoustic parameters that we did not analyse, such as amplitude modulation (Fichtel & Hammerschmidt, 2001). Our recordings were made under natural conditions with free-ranging animals, which led to various constraints, such as recording at varying distances, directions and atmospheric conditions, which made it impossible to derive reliable measures of amplitude modulation. Future research on peeps using a larger sample size in more controlled acoustic conditions could address this issue. More conclusive evidence would require carefully designed playback experiments to test whether bonobos have more difficulties discriminating positive and neutral peeps, compared to negative peeps. This is important because the notion of functional flexibility also makes predictions about signaller-receiver interactions, i.e., between a signaller’s communicative act, or ‘illocutionary force,’ and a recipient’s consequential response, or ‘perlocutionary effect.’ For example, an infant’s “complaints” may be associated with specific vocalisations showing negative affect, which then induces a caregiver offering to feed after interpreting the negative affect vocalisation as an indicator of hunger (Austin, 1975; Bates et al., 1979; Oller et al., 2013). The interaction between signaller and receiver would be a relevant aspect to explore in future work, such as using match to sample experiments or naturalistic playback studies.

While functional flexibility is undoubtedly more developed in human speech, results from the current study contribute to the debate surrounding speech evolution by highlighting evidence for a possible intermediate stage in the communicative repertoire of our closest living relative, the bonobo. While much of bonobo vocal behaviour is functionally fixed by nature, such as screaming during agonistic encounters, barking during alarm and laughing during play (de Waal, 1988), their most common vocalisation, the ‘peep’ is also produced in highly flexible ways across a wide range of contexts and valence states. Previously, the peep vocalisation had been primarily considered as a food-associated call that was combined with other calls in a sequence in order to convey information about food quality (de Waal, 1988; Clay & Zuberbühler, 2009; Clay & Zuberbühler, 2011). It is possible that when combined in a particular way with other calls, or when combined with a certain context, peeps may still functionally refer to specific events in the external world (Scarantino & Clay, 2014). Future research, probing the manner in which peeps are combined with other calls and contextual stimuli, is needed to address this question.

It is relevant to note that studies of language-competent bonobos have already highlighted bonobos as a good model for studying the prerequisites for language evolution (Hopkins & Savage-Rumbaugh, 1991; Taglialatela & Savage-Rumbaugh, 2003). In addition to their remarkable ability to acquire human and artificial language systems in captivity (e.g., Segerdahl, Fields & Savage-Rumbaugh, 2005), bonobo vocal behaviour has been suggested to be more flexible and dialogue-like than chimpanzees. An interesting possibility is that peeps function to draw attention to and “comment” on novel items or environmental events (de Waal, 1988; Savage-Rumbaugh, Shanker & Taylor, 1998), a communicative behaviour also found in early language development (Tomasello & Carpenter, 2007). Future research should determine whether bonobos are also capable of varying the expression of emotional valence across different utterances within the same vocal event using the same vocal type.

An alternative possibility is that the production of peeps fits within a more generalised function that extends across multiple contexts (Notman & Rendall, 2005). For instance, it may have a broader social function related to cohesion or spacing which may explain patterns of usage across seemingly disparate behavioural contexts. For example, the vocalisations produced during feeding by dolphins (Janik, 2000) and greater-spear nosed bats (Wilkinson & Boughman, 1998) have been suggested to function to coordinate social foraging rather than being specific to feeding per se. The rationale for cohesion effects may even be supported by the current results regarding negative contexts, since cohesion in this case could be outweighed by the urge to escape a predator or a conspecific.

Another proximate explanation for shared acoustic structure is an underlying a shared motivational aspect to the eliciting contexts in question. To address this issue, a solid framework to assess expressions of affect in great apes is first required. One approach, already employed for human infants (Oller et al., 2013), could involve comparing affect expression in corresponding modalities, such as coding facial expressions produced in association with the vocalisation (e.g., Parr, Waller & Heintz, 2008; Parr, Waller & Vick, 2007). However, unlike the facial affect analyses conducted in association with vocalisation production for human infants (Oller et al., 2013), bonobo peep vocalisations are closed-mouth vocalisations, making facial affect coding less appropriate. Future work using associated playbacks and hormonal analyses may also provide relevant insights into underlying motivational states during vocal production (Mateo, 2010).

In summary, the current study contributes promising insights into the evolution of human speech by suggesting an intermediate stage between fully-fledged functional flexibility in human speech production and the more traditionally viewed of fixed signals of non-human primates. Rather than being a uniquely human trait, results from the current study reflect a growing body of literature that suggests that flexible vocal signalling (e.g., Lemasson, Hausberger & Zuberbühler, 2005; Lemasson et al., 2011; Ouattara, Lemasson & Zuberbühler, 2009; Crockford et al., 2012; Koda et al., 2013; Liebal et al., 2013) and perhaps even functional arbitrariness (Lameira et al., 2013) may have deeper roots in the primate lineage than previously assumed. By demonstrating the potential for functional flexibility in the vocal behaviour of a great ape species, the results provide a useful springboard from which evidence for functional flexibility in other animal species and across different modalities can be explored.

Supplemental Information

File S1 Bonobo peep vocalisation produced by male ZD during feeding

Click here for additional data file.

File S2 Bonobo peep vocalisation produced by male ZD in response to alarm

Click here for additional data file.

File S3 Bonobo peep vocalisation produced by male ZD during rest

Click here for additional data file.

We thank Gottfried Hohmann for granting us permission to conduct research at Lui Kotale and for ongoing support and advice. We are grateful to the Institut Congolaise pour la Conservation de la Nature (ICCN) for granting permission to conduct research at Salonga National Park (MIN.0242/ICCN/DG/GMA/013/2013). The methods used to collect observational data in the field are in compliance with the requirements and guidelines of the ICCN and adhere to the legal requirements of the host country, the Democratic Republic of Congo. We are very grateful to Isaac Schamberg for his support in the field, to Barbara Fruth for her collaboration in supporting our research at Lui Kotale Bonobo Project and to members of Lompole village and all local staff supporting Lui Kotale. We are grateful to all members of the Department of Comparative Cognition at the University of Neuchatel for stimulating discussion, and especially to Christof Neumann for on-going support and advice. We thank Jennifer Vonk, Adriano Lameria, Heidi Lynn and an anonymous reviewer for their highly insightful comments on the manuscript.

Additional Information and Declarations

Competing Interests

Author Contributions

Animal Ethics

Data Availability

The authors declare there are no competing interests.

Zanna Clay conceived and designed the experiments, performed the experiments, analyzed the data, contributed reagents/materials/analysis tools, wrote the paper, prepared figures and/or tables, reviewed drafts of the paper.

Jahmaira Archbold analyzed the data.

Klaus Zuberbühler conceived and designed the experiments, contributed reagents/materials/analysis tools, wrote the paper, reviewed drafts of the paper.

The following information was supplied relating to ethical approvals (i.e., approving body and any reference numbers):

Institut Congolais pour la Conservation de la Nature provided full approval for this purely observational research (MIN.0242/ICCN/DG/GMA/013/2013).

The following information was supplied regarding the deposition of related data:

Figshare: figshare.com/s/828f0bac209811e585d106ec4b8d1f61.

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
