# Peer review of "Functional flexibility in wild bonobo vocal behaviour"

_PeerJ, doi:10.7717/peerj.1124_

## Round 0.1 · original submission · Minor Revisions

· Academic Editor

Minor Revisions

The three expert reviewers have all identified great value in your contribution and indeed encourage you to highlight the critical advances from your study. At the same time they ask you to be circumspect given the limitations that you have carefully acknowledged. Admittedly, this is a bit of a balancing act, but one I am confident you can manage. The reviewers have been extremely thorough in their comments and suggestions (please see the attached review from Reviewer 2 for detailed notes). I advise you to provide all of the requested clarifications to increase the impact of your work. Supplementary files including call samples would be welcome, although are not required for your revision.

·

Basic reporting

Clay, Archbold and Zuberbühler present a study on peep use and potential acoustic variation across contexts in wild bonobos. This study focuses on the least known species of Pan and on its most commonly produced vocalization type, even if relatively little is actually known about it currently. In particular, the study focuses on bonobo peep behavior in natural conditions, constituting therefore a fundamental contribution to the understanding of the bonobo vocal behavior as a whole, and a desirable milepost paper both for past studies on this vocalization type in captivity, as well as any future study on bonobo calls.

The paper is very well written and presents a clear line of thought throughout. The explanation of the state of art is very comprehensive, being extremely complete with regards to referencing and contextualizing previous studies in light of the aims of this particular study. The methods are well explained and statistically sound, demonstrating how “routine” acoustic analyses, such as on individual and contextual variation, can be used as powerful tool to investigate new aspects of great ape vocal behavior.

I enthusiastically recommend this paper for publication in PeerJ, and I present only minor suggestions which the authors may consider including for the matter of clarity.

Adriano Lameira

Experimental design

No comments.

Validity of the findings

No comments.

Additional comments

Line 24: I would specify here that the differences were specifically absent between positive and neutral peeps.

207-215: This section is slightly confusing. I suggest rephrasing and/or reducing the text. For instance, 209-212, simply indicate “We collected peeps at the onset of each context”. After re-reading this section, I understand that, though you decided to measure the first 3 calls, these were not always available, with individuals sometimes only producing 1 or two calls at the onset of the context. Is this correct? Was there any specific reason to consider maximal 3 calls? This doesn’t need to be very technical reason, simply to understand if or how this was due to the calling dynamics of bonobos.

226: Perhaps insert here the details about the calls of negative valence, instead of being presented down in the text in 233. Otherwise, move 226-229 down in the text.

246: Does the script have a name, or is it available online? Was the script, by any chance, part of the GSU Praat tool package developed by Owren?

299-307: This is really remarkable! Though this section is simply descriptive, it transmits in a formidable and potent way the extent of function flexibility of bonobo peeps. Even though the authors do not compare acoustically all these contexts (naturally due to statistical limitations), the observation that the same call type is produced across more than 12 distinct contexts is simply astonishing.

310: omit comma after “contexts”

330: simply state “peeps associated with negative valence” since all these were from non-feeding contexts. It may cause confusion to the reader in the sense that it may seem implicit from the text that there were also “peeps associated with feeding contexts with negative valence”.

378: omit “the”

390: change “acoustic” into “contextual”

405: add full stop before “Currently”.

395-408: I would also take some time here to discuss the potential reasons for the acoustic differences that the peeps of negative valence present. I suggest this because, comparatively to positive and neutral peeps, negative peeps were higher pitched (also at call onset) and shorter. These are both direct effects of higher subglottal air pressures during call production. Considering theoretically that peep production is associated with a specific vocal fold positioning and corresponding air volume that excites the vocal folds (we omit any dimensions of supralaryngeal position for peep production since formants were not investigated here), one may in fact predict that peeps will be higher pitched and shorter in negative contexts, as these are charged, tense and urgent contexts requiring the immediate behavioral responses from individuals. The same volume of air passes at higher speed at the larynx, inciting vocal fold oscillation at a higher frequency for a shorter period of time. This “formula” for peep production is naturally simplistic, but it could be useful for several reasons along the discussion lines of the authors. First, the authors forward that bonobo functional flexibility meets human’s half way. It could be relevant, from an evolutionary point, to speculate in light of these results, that functional flexibility could have occurred first between positive and neutral contexts in the course of language evolution, and that the physical mechanics of vocal production make it naturally hard to conceal negative valence. Second, as the authors clearly state, acoustic analyses were not (!) conducted in the study demonstrating functional flexibility in human children. If the above “formula” is correct, then one predicts it will apply to vocal fold action both in great apes and humans. Thus, that even in children negative valence will be perspicuous.

425: I would avoid here the use of the term “dimensions” (perhaps “domains” instead?) since you are explaining the “dimensional approach”.

434: omit “is”

437: Start new paragraph.

442: “Facial expressions” or “affect states”

447: I would bring in here whether peeps were produced with constant vs. varying and neutral vs. significant facial expressions.

449-457: I think that the authors may be perhaps too stringent on themselves in this section. Personally, I have attempted to include call amplitude measures in several studies, and it is a point which regularly brings questions from reviewers simply because these frequently question the reliability of this parameter’s measurements, even when observers took all effort to control for the factors that, as the authors mention, are known influence amplitude measures. The inclusion of amplitude measures usually also reduces the sample of available calls that may be used for analyses (in the sense that only calls produced in the same bout with similar recordings conditions may be comparable). Since some of the exciting prospects of this study for future research includes the expansion of the sample sizes available, altogether, I think may be unwarranted to bring up this matter on amplitude measures.

459-473: Since the discussion allocates now substantial space to topics that do not directly derive from the results, the authors may consider omitting this paragraph.

480: insert comma before “barking”

501-504: I do not fully understand the rationale for this statement; wasn’t this essentially what the authors did in the present study?
In the note from the authors to the Editor, the authors state that individuals likely rely on information other than the acoustic to extract call function and meaning since peeps do not present contextual differences between positive and neutral contexts. I believe this is perhaps something worth dedicating some attention to in the discussion.

Reviewer 2 ·

Basic reporting

Please see file attached.

Experimental design

Please see file attached.

Validity of the findings

Please see file attached.

Additional comments

Please see file attached

Annotated reviews are not available for download in order to protect the identity of reviewers who chose to remain anonymous.

·

Basic reporting

Article seems to adhere to all policies. My only suggestions for the clarity of writing are minor (see below).

Experimental design

While I am not an expert on Discriminate Function Analysis, the analysis seems appropriate and solid. Possible confounds due to wild data collection are all addressed appropriately.

Validity of the findings

Findings seems valid and are congruent with current thinking with regard to vocal flexibility in nonhuman primates. In general, I think it needs to be made clearer from the outset that the peep vocalizations seem to be flexible across some contexts, but not across others. Also, i got a little lost with the comparisons to the infant data, some clarification would be greatly appreciated.

Additional comments

minor issues and points of clarification/curiosity:
Line 174 – not clear to the reader how this differs from the discrete emotions approach. Do you mean you are not naming individual emotions? Maybe an example of the discrete emotions would help clarify. And why is this approach better for comparisons? Clarify, please.
Line 223 – this would be much easier to understand if you presented all three valences and contexts in the same place. (Negative valences not mentioned until line 235)
Line 299 – this seems like it belongs more in the methods section
Line 415 – no mention of the arousal dimension before this – was this used in the analysis?
Line 511 – Janik is spelled wrong (not Janick)
Slocombe recent work?
I think it would be helpful in the text surrounding figure 2 to mention the basic findings – that positive and neural valence vocalizations did not differ on these measures, but negative valence did.
I’d also like some mention of possible explanations for the three way comparison not showing significance – was it just the similarity of the positive and neutral valence vocalizations?
I’m curious about the outlier vocalizations, was there something different about those contexts? OR were they from a particular individual? Did you calculate the z-scores based on individual vocalization ranges or the group?
What were the measurements that allowed for individual identification? Is this the section that starts at line 357? Or does that need another subheading?

---

## Round 0.2 · Minor Revisions

· Academic Editor

Minor Revisions

I have only a few very minor suggestions that I would like for you to make before I can officially accept the paper.

I would delete the clause “representing the principle of human speech evolution” from the end of the abstract.

I don’t actually follow the link between the first two sentences of the introduction. Why does human development speak to the idea of adding complexity across evolution? Please make more explicit.

On line 86, should this be “Gorilla gorilla gorilla” to denote western lowland gorilla?

Perhaps you could say something more in the introduction about why you focused exclusively on peeps (as opposed to any other bonobo vocalizations). If these are indeed the most frequently and indiscriminately used vocalizations it might seem like “stacking the deck” so to speak; however, you might argue that it makes sense to look for a phenomenon in the case most likely to find it. I’d simply like to see an objective justification for this focus stated explicitly.

On line 243 could you rephrase to eliminate the repetition of “using”.
Write out in full discriminant function analysis (DFA) the first time it appears (line 274) rather than on line 285. Then you don’t need to write it out in full again (line 322).

Standardized is spelled incorrectly on line 299.

Thank you again for submitting such an interesting paper to PeerJ.

---

## Round 0.3 · accepted · Accept

· Academic Editor

Accept

Thank you for being prompt and responsive to the last round of requested revisions. I am now pleased to accept your MS for publication in PeerJ. Thank you for submitting such an interesting paper.